# Retrieval-Generation Alignment for End-to-End Task-Oriented Dialogue System

**Weizhou Shen[1], Yingqi Gao[1], Canbin Huang[1], Fanqi Wan[1], Xiaojun Quan[1]\*, Wei Bi[2]\***

[1]School of Computer Science and Engineering, Sun Yat-sen University, China
[2]Tencent AI Lab
{shenwzh3, gaoyq28, huangcb3, wanfq}@mail2.sysu.edu.cn,
quanxj3@mail.sysu.edu.cn, victoriabi@tencent.com

## Abstract

Developing an efficient retriever to retrieve knowledge from a large-scale knowledge base (KB) is critical for task-oriented dialogue systems to effectively handle localized and specialized tasks. However, widely used generative models such as T5 and ChatGPT often struggle to differentiate subtle differences among the retrieved KB records when generating responses, resulting in suboptimal quality of generated responses. In this paper, we propose the application of maximal marginal likelihood to train a perceptive retriever by utilizing signals from response generation for supervision. In addition, our approach goes beyond considering solely retrieved entities and incorporates various meta knowledge to guide the generator, thus improving the utilization of knowledge. We evaluate our approach on three task-oriented dialogue datasets using T5 and ChatGPT as the backbone models. The results demonstrate that when combined with meta knowledge, the response generator can effectively leverage high-quality knowledge records from the retriever and enhance the quality of generated responses. The code of this work is available at https://github.com/shenwzh3/MK-TOD.

## 1 Introduction

Task-oriented dialogue systems (TOD) assist users to accomplish daily tasks such as restaurants, scheduling appointments, and navigating traffic by leveraging external knowledge bases. Among them, pipeline systems (Henderson et al., 2014; Hosseini-Asl et al., 2020) involve several intermediate stages such as dialog state tracking and system policy learning for retrieving knowledge and generating responses. In contrast, end-to-end task-oriented dialog systems (E2E-TOD) (Wu et al., 2022; Tian et al., 2022) have gained increasing concentration for their ability to directly generate responses based on the knowledge base without intermediate annotations. Although the end-to-end paradigm appears

---
*Corresponding authors.

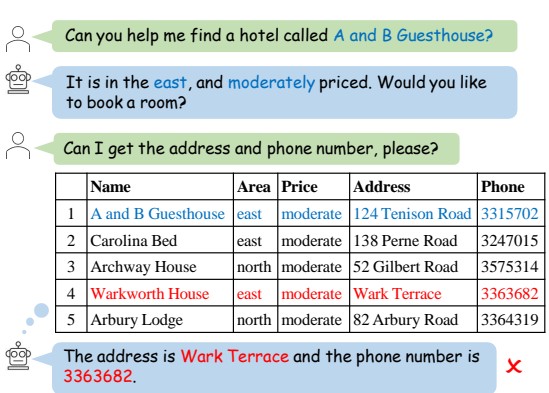

Figure 1: A demonstrative case in E2E-TOD. The table displays retrieved entities sorted by retrieval order. The correct entity is highlighted in blue. However, the response generator mistakenly selects the false entity, highlighted in red, leading to an erroneous response.

| | Name | Area | Price | Address | Phone |
|---|---|---|---|---|---|
| 1 | A and B Guesthouse | east | moderate | 124 Tenison Road | 3315702 |
| 2 | Carolina Bed | east | moderate | 138 Perne Road | 3247015 |
| 3 | Archway House | north | moderate | 52 Gilbert Road | 3575314 |
| 4 | Warkworth House | east | moderate | Wark Terrace | 3363682 |
| 5 | Arbury Lodge | north | moderate | 82 Arbury Road | 3364319 |

to be more compatible with practical scenarios and large-scale language models, it imposes challenges in acquiring and utilizing external knowledge as no belief state is provided for knowledge retrieval.

Retrieval-augmented generation (Lewis et al., 2020; Ren et al., 2021; Singh et al., 2021) has demonstrated success in various knowledge-intensive tasks by employing a held-out dense retriever to retrieve knowledge and then taking the knowledge to generate results. Q-TOD (Tian et al., 2022) applies this approach to E2E-TOD and significantly outperforms previous methods that combine knowledge retrieval and response generation into a single model (Madotto et al., 2018; Qin et al., 2020; Raghu et al., 2021). However, our preliminary study (Section 5.3) shows that under this framework the correlation between the performance of knowledge retriever and that of response generator is relatively weak, meaning that simply improving the retriever may not lead to a better generator. We characterize this phenomenon as the *misalignment* between the retrieval and generation processes in E2E-TOD systems. This misalignment poses a bottleneck for current dialogue systems, as improve-

ments in the retriever component do not necessarily translate to enhanced generation quality.

Through qualitative analysis, we hypothesize that the misalignment between retrieval and generation is attributed to the homogeneity of retrieved knowledge entities. As illustrated in Figure 1, the retrieved entities exhibit a high degree of similarity, with only minor variations in their values. Consequently, since the response generator is trained on reference responses that predominantly consist of language tokens rather than knowledge-related tokens, it struggles to differentiate between similar entities and may inadvertently select inappropriate entities for response generation.

In this paper, we introduce **M**eta **K**nowledge for end-to-end **T**ask-**O**riented **D**ialogue system (MK-TOD) as a solution to address the retrieval-generation misalignment. MK-TOD aims to correlate the performance of the knowledge retriever and response generator for improved system performance. To enhance the knowledge retriever, we propose the application of maximum marginal likelihood (Singh et al., 2021) for progressive retriever updating during the training of the response generator. In order to enable the response generator to distinguish between entities, we explore several methods for utilizing retrieval-related *meta knowledge*. Here, meta knowledge refers to various information about the retrieved entities, such as retrieval order, retrieval confidence, and co-occurrence rate. We propose three approaches for incorporating the meta knowledge: adding special prefix tokens, using prompts, and applying contrastive learning. Additionally, we investigate the introduction of negative knowledge during the generator's training to enhance its discriminative ability.

We apply MK-TOD to several backbone models, including T5 (Raffel et al., 2020) and the large language model ChatGPT (OpenAI, 2022). We compare MK-TOD with other E2E-TOD systems on three benchmark datasets, namely SMD, CamRest, and WoZ (Eric et al., 2017; Wen et al., 2017; Eric et al., 2020). The empirical results demonstrate the superiority of our proposed system over the current state-of-the-art systems with similar model scales. Additionally, our system effectively enhances the performance of ChatGPT in E2E-TOD with in-context learning. Furthermore, through comprehensive analysis, we uncover that our meta-knowledge approach successfully alleviates the misalignment between the retriever and generator. This approach empowers the generator to better differentiate between similar entities during response generation.

## 2 Related Works

### 2.1 End-to-End Task-Oriented Dialogue

The existing work on the usage of external knowledge in end-to-end task-oriented dialogue systems can be divided into three categories. The first category takes the whole knowledge base as the model input, and conducts knowledge selection and response generation in one single model. For instance, Mem2seq (Madotto et al., 2018), KB-Retriever (Qin et al., 2019), GLMP (Wu et al., 2019) and CDNET (Raghu et al., 2021) employ memory networks for querying knowledge. UnifiedSKG (Xie et al., 2022) directly concatenates entities as the input of Transformers. The second category directly encodes knowledge into model parameters. GPT-KE (Madotto et al., 2020) pretrains their model on augmented dialog data to embed the knowledge base, while ECO (Huang et al., 2022) applies tri-constraints on top of GPT-KE to ensure entity consistency. The third category is to use an individual retriever to retrieve knowledge. For example, Q-TOD (Tian et al., 2022) decouples the dialogue system into a retriever and a generator and uses the generator to generate a query for knowledge retrieval. DialogKG (Rony et al., 2022) inputs the flattened records to a graph neural network to select entities. MAKER (Wan et al., 2023) introduces a multi-grained retriever with both entity and attribute selection. As mentioned earlier, although the retrieve-then-generate framework has been a successful paradigm to date, it leads to misalignment between the retriever and the generator in end-to-end task-oriented dialogue systems.

### 2.2 Retrieval-Augmented Generation

With the success of the dual-encoder neural retriever (Karpukhin et al., 2020), the retrieval-augmented generation framework is widely applied to various knowledge-intensive tasks. This framework uses a retriever to retrieve knowledge from a knowledge base and inputs the retrieval results into a generator to generate the answer. Among them, RAG (Lewis et al., 2020) separately encodes each retrieved knowledge record with the query and marginalizes the probabilities of the answer based on each entity. FiD (Izacard and Grave, 2021) encodes each retrieved knowledge like RAG and fuses their hidden states in the decoder. FiD-

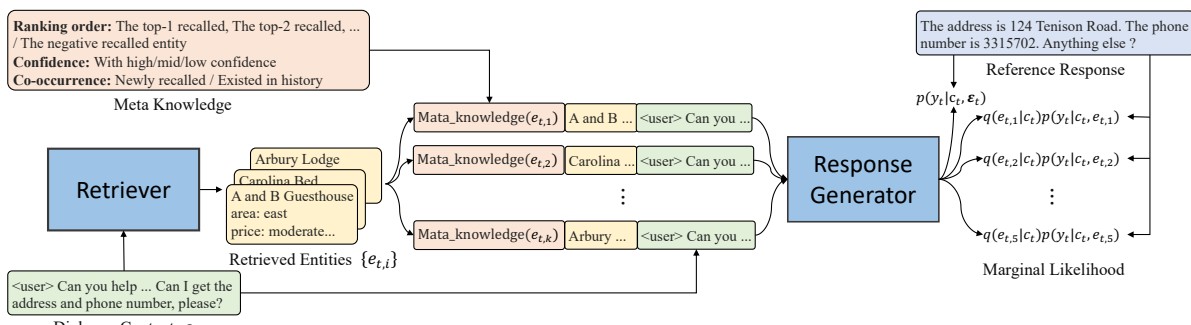

Figure 2: The MK-TOD framework comprises a knowledge retriever and a response generator. Given the dialogue context, the retriever retrieves entities from the knowledge base. Each entity is concatenated with its corresponding meta knowledge and subsequently input into the generator to generate the response. The optimization process involves two likelihoods: the normal text generation likelihood and the marginal likelihood.

KD (Izacard and Grave, 2022) and EMDR$^2$ (Singh et al., 2021) are both based on the FiD framework but with different retriever training methods: FID-KD uses knowledge distillation while EMDR$^2$ uses marginal likelihood. REPLUG (Shi et al., 2023) applies the method of RAG to large language models but only updates the retriever during training.

## 3 Methodology

The framework of our proposed MK-TOD is depicted in Figure 2. It consists of a retriever and a response generator. In each dialogue turn, the retriever retrieves a set of relevant entities, which are then combined with retrieval-related meta knowledge and the dialogue context. The generator utilizes this information to generate a response for the current turn. In the following section, we first introduce the notations and provide an overview of our method. Then, we delve into the detailed explanations of two crucial components: maximum marginal likelihood and meta knowledge.

### 3.1 Notations

Given a dialogue $\mathcal{D} = \{u_1, r_1, ..., u_T, r_T\}$ of $T$ turns, where $u_t$ and $r_t$ are the $t$-th turn user utterance and system response, respectively. We use $c_t$ to represent the dialog context of the $t$-th turn, where $c_t = \{u_1, r_1, ..., u_{t-1}, r_{t-1}, u_t\}$. An external knowledge base (KB) is provided in the form of a set of entities, i.e., $\mathcal{K} = \{e_1, e_2, ..., e_B\}$, where each entity $e_i$ consists of several attribute-value pairs and $B$ is the size of knowledge base. End-to-end task-oriented dialog systems take dialogue context $c_t$ and knowledge base $\mathcal{K}$ as input and generate an informative natural language response $r_t$.

### 3.2 System Overview

The retriever module comprises a context encoder and an entity encoder. The context encoder transforms the current dialogue context $c_t$ into a vector representation $h_{c_t}$. On the other hand, the entity encoder concatenates the attribute-value pairs of each entity as plain text and encodes it into a vector representation $h_{e_i}$. The matching score $s_{t,i}$ is computed by taking the dot product between $h_{c_t}$ and $h_{e_i}$. Consequently, the top-$K$ entities with the highest scores are selected as candidate entities $\mathcal{E}_t$ for the current dialogue turn. Furthermore, if meta knowledge is utilized, each entity in $\mathcal{E}_t$ is augmented with its corresponding meta knowledge.

The generator takes the retrieved entities $\mathcal{E}_t$ and dialogue context $c_t$ as inputs to generate the final system response $r_t$. The probability of generating the response $r_t$ given the entities $\mathcal{E}_t$ and dialogue context $c_t$ can be calculated as follows:

$$p(r_t|c_t, \mathcal{E}_t; \theta) = \prod_{j=1}^{|r_t|} p(r_{t,j}|r_{t,<j}, c_t, \mathcal{E}_t; \theta), \quad (1)$$

where $\theta$ denotes the parameters of the generator.

Similar to most text generation tasks, we incorporate the negative log-likelihood (NLL) loss as a training objective to train the generator:

$$\mathcal{L}_{NLL} = -\log p(r_t|c_t, \mathcal{E}_t; \theta). \quad (2)$$

### 3.3 Maximum Marginal Likelihood

Due to the absence of retrieval labels for training the retriever, we depend on supervision signals from the generator. However, since it is not possible to backpropagate gradients through the NLL loss in Eq. (2) to the retriever, we propose updating the retriever's parameters by maximizing the

marginal likelihood (MML) of the response $r_t$. The marginal likelihood offers a Bayesian perspective to compute $p(r_t|c_t, \mathcal{K})$ by integrating the likelihood over all the entities in the knowledge base:

$$p(r_t|c_t; \phi, \theta) = \sum_{e_i \in \mathcal{K}} q(e_i|c_t; \phi) p(r_t|c_t, e_i; \theta),$$

(3)

where $\phi$ denotes the parameters of the retriever and $q(e_i|c_t; \phi)$ is the retrieval probability of entity $e_i$. Note that computing $q(e_i|c_t; \phi)$ for all entities in the entire knowledge base would incur an unaffordable computational cost for Eq. (3). Therefore, following the approach of EMDR[2] (Singh et al., 2021), we compute $q(e_i|c_t; \phi)$ over the retrieved entities $\mathcal{E}_t$ instead of the entire knowledge base $\mathcal{K}$:

$$p(r_t|c_t; \phi, \theta) = \sum_{e_{t,i} \in \mathcal{E}_t} q(e_{t,i}|c_t; \phi) p(r_t|c_t, e_{t,i}; \theta),$$

(4)

where $q(e_{t,i}|c_t; \phi)$ is implemented as follows:

$$q(e_{t,i}|c_t; \phi) = \frac{\exp(s_{t,i})}{\sum_{e_{t,j} \in \mathcal{E}_t} \exp(s_{t,j})}.$$

(5)

The loss function for the marginal likelihood is defined as follows:

$$\mathcal{L}_{MML} = -\log \sum_{e_{t,i} \in \mathcal{E}_t} q(e_{t,i}|c_t; \phi) p(r_t|c_t, e_{t,i}; \theta).$$

(6)

By incorporating $q(e_{t,j}|c_t; \phi)$, we can propagate gradients back to the retriever and update its parameters. The ultimate training loss function for MK-TOD is defined as follows:

$$\mathcal{L} = \alpha \mathcal{L}_{NLL} + \beta \mathcal{L}_{MML},$$

(7)

where $\alpha$ and $\beta$ are hyperparameters.

## 3.4 Meta Knowledge

We introduce the concept of retrieval-related meta knowledge, which encompasses various information about the retrieved entities to guide the generator and enhance the alignment between retrieval and generation. Three key factors are considered in the meta knowledge: retrieval order, retrieval confidence, and co-occurrence.

**Retrieval order**: The retriever evaluates entities based on their matching with the dialogue context, prioritizing those that exhibit a higher degree of matching. We incorporate the retrieval order of each entity as a part of the meta knowledge.

**Retrieval confidence**: To provide more retrieval information, we categorize retrieved entities into low-confidence, middle-confidence, and high-confidence based on retrieval scores. The thresholds for categorizing entities are hyper-parameters[1]. Retrieval confidence, in conjunction with retrieval order, enables the generator to disregard entities with low confidence but high retrieval order.

**Co-occurrence**: Entities that have already appeared in the dialogue context are more likely to be relevant for future responses. Thus, we inform the generator about the occurrence of entities in the dialogue context through meta knowledge.

To implement the above meta knowledge in our system, we design three approaches: prefix, prompt, and contrastive learning.

### 3.4.1 Prefix

In this approach, we create a mapping function that assigns special tokens representing meta knowledge to each entity. For instance, an entity ranked second in retrieval order, with middle retrieval confidence, and not yet mentioned in the context would be mapped to the set of `<2nd-entity>`, `<mid-confidence>`, `<new-entity>`[2]. These prefix tokens are then concatenated with the corresponding entity and input to the generator during both training and inference stages.

### 3.4.2 Prompt

To fully leverage the generator's language modeling capability, we explore using prompts to incorporate meta knowledge. Here, we design a mapping function that assigns each entity a set of prompts, which are natural language sentences representing the meta knowledge. For example, a prompt can be "This is the top-1 recalled entity with low confidence".[3] Similar to the prefix approach, these prompts are concatenated with the corresponding entities and fed into the generator.

### 3.4.3 Contrastive Learning

We can also train the generator to distinguish between entities by employing contrastive learning. In this approach, we select a subset of entities from the retrieved entities $\mathcal{E}_t$ based on their retrieval or-

---

[1]We established the following ranges: (-infinity, 0.4] indicates low confidence, (0.4, 0.75] indicates medium confidence, and (0.75, +infinity) indicates high confidence.
[2]The complete mapping is discussed in Appendix B.1
[3]The complete mapping is discussed in Appendix B.2

der, forming a positive entity set $\mathcal{E}_t^*$.[4] For each entity $e_{t,i}$ in $\mathcal{E}_t^*$, we compute its length-normalized log-likelihood of generating the response:

$$d_{t,i} = \frac{\log p(r_t|c_t, e_{t,i}; \theta)}{|r_t|}, \quad (8)$$

where $|r_t|$ is the length of $r_t$. Additionally, we calculate the log-likelihood of generating the response without any entity as the baseline likelihood:

$$d_t^- = \frac{\log p(r_t|c_t; \theta)}{|r_t|}. \quad (9)$$

We employ a pairwise marginal ranking loss that ensures the likelihood of positive entities greater than the baseline likelihood by a certain margin:

$$\mathcal{L}_{CTR} = \sum_{e_{t,i} \in \mathcal{E}_t^*} \max(0, d_t^- - d_{t,i} + \lambda), \quad (10)$$

where $\lambda$ is a hyperparameter. We then add this loss term to the loss function of MK-TOD:

$$\mathcal{L} = \alpha \mathcal{L}_{NLL} + \beta \mathcal{L}_{MML} + \gamma \mathcal{L}_{CTR}. \quad (11)$$

### 3.5 Negative Entity

Inspired by negative sampling in information retrieval (Karpukhin et al., 2020), we also consider incorporating negative entities into the generator. The negative entity, denoted as $e_t^- \notin \mathcal{E}_t$, is chosen as the entity with the lowest retrieval score from $\mathcal{K}$. Special meta knowledge is designed for the negative entity as well.[5] Note that the negative entity is different from the baseline likelihood in the above contrastive learning (Section 3.4.3).

### 3.6 Model Inference

During inference, we first retrieve entities using the retriever. Then, we prepend each entity with its corresponding meta knowledge. Finally, we concatenate the entities with the dialogue context and input the resulting sequence to the generator to generate the final response. Notably, we do not include negative entities during inference.

### 3.7 Discussion

The motivation of meta knowledge is to empower the generator by enabling it to utilize retrieved information for improving the generation process.

---

[4]Retrieval confidence and co-occurrence are not considered to avoid sparsity in the positive entity set.

[5]Details are provided in Appendix B.1 and B.2

It's important to highlight that, while we intentionally provide the generator with meta knowledge, such as retrieval order and confidence scores, this does not compel the generator to exclusively rely on the highest-ranked entities. Rather, our approach encourages the generators to autonomously distinguish between these entities.

## 4 Experimental Settings

### 4.1 Datasets

We evaluate our MK-TOD on three task-oriented dialogue datasets: MultiWOZ 2.1 (MWOZ) (Eric et al., 2020), CamRest (Wen et al., 2017), and Stanford Multi-Domain (SMD) (Eric et al., 2017). We compare the methods with two different settings about the knowledge bases: First, each dialogue has a small session-level knowledge base associated with the user goal, which is the typical setting of previous work. Second, all conversations in MWOZ/CamRest share a dataset-level large-scale knowledge base by injecting all the session-level knowledge bases. There are 223 and 112 entities in the large-scale knowledge base for MWOZ and CamRest, respectively. Other detailed statistics of these datasets are shown in Appendix A.

For all three datasets, we employ BLEU (Papineni et al., 2002) and Entity F1 (Eric et al., 2017) as the metrics to evaluate the quality of generated responses. Entity F1 assesses the presence of accurate knowledge in the responses by calculating the micro-averaged precision and recall scores of attribute values. Additionally, for experiments conducted on large-scale knowledge bases, we introduce Recall@K as the performance metric for the retriever. Recall@K measures the percentage of gold entities appearing in the retrieved entities.

### 4.2 Implementation Details

We utilize BERT (Devlin et al., 2019) as the context encoder and entity encoder for the retriever. As for the generator, we employ T5 (Raffel et al., 2020) and ChatGPT (OpenAI, 2022). Note that ChatGPT is not fine-tuned but instead undergoes in-context learning using our datasets.[6] The retriever for ChatGPT is directly copied from the retriever trained with T5 using MML. All experiments are performed on a single 24G NVIDIA RTX 3090 GPU, and the best checkpoints are selected based on Entity F1 scores on the validation set. Hyperparameter settings are listed in Appendix E.

---

[6]Further details are provided in Appendix C

| Model | MWOZ | | | CamRest | | |
|---|---|---|---|---|---|---|
| | BLEU | Entity F1 | Recall@7 | BLEU | Entity F1 | Recall@7 |
| DF-Net (Qin et al., 2020) | 6.45 | 27.31 | - | - | - | - |
| EER (He et al., 2020b) | 11.60 | 31.86 | - | 20.61 | 57.59 | - |
| FG2Seq (He et al., 2020a) | 10.74 | 33.68 | - | 19.20 | 59.35 | - |
| CDNET (Raghu et al., 2021) | 10.90 | 31.40 | - | 16.50 | 63.60 | - |
| Q-TOD (T5-Large) (Tian et al., 2022) | 15.52 | 46.74 | 92.97 | 21.44 | 63.88 | **95.52** |
| MAKER (T5-Base) (Wan et al., 2023) | 16.25 | 50.87 | - | 26.19 | 72.09 | - |
| MAKER (T5-Large) (Wan et al., 2023) | 18.23 | 52.12 | - | 25.34 | 72.43 | - |
| Ours$_{prefix}$ (Base) | 16.39 | 50.35 | 92.51 | 25.23 | 71.15 | 94.35 |
| Ours$_{prompt}$ (Base) | **17.56** | 50.69 | 92.74 | 26.69 | 71.67 | 92.24 |
| Ours$_{ctr}$ (Base) | 15.96 | 51.35 | 92.74 | 26.85 | **73.51** | 93.88 |
| Ours$_{prefix}$ (Large) | 16.69 | **53.59** | 92.93 | 27.32 | 72.77 | 91.08 |
| Ours$_{prompt}$ (Large) | 17.15 | 52.99 | 94.42 | 26.88 | 72.92 | 95.41 |
| Ours$_{ctr}$ (Large) | 17.40 | 53.26 | **95.22** | **27.82** | 71.98 | 95.38 |
| ChatGPT (OpenAI, 2022) | 6.79 | 30.31 | 92.74* | 14.76 | 52.92 | 94.35* |
| Ours$_{prefix}$ (LLM) | 7.01 | 30.69 | 92.74* | 14.51 | 52.38 | 94.35* |
| Ours$_{prompt}$ (LLM) | **7.31** | **32.04** | 92.74* | **14.91** | **53.58** | 94.35* |

Table 1: Overall results of E2E-TOD systems with large-scale knowledge bases on MWOZ and CamRest, where "*" means that we directly use the retriever co-trained with T5-Base using MML.

Consistent with previous studies (Qin et al., 2019), we initialize the retriever through pre-training with distant supervision to prevent collapsed representations. Additional details on the pre-training process can be found in Appendix D.

## 4.3 Baseline Methods

We include several strong baselines for comparison.
**Implicit retrieval**: These methods combine knowledge retrieval and response generation in a single model, including DF-Net (Qin et al., 2020), EER (He et al., 2020b), FG2Seq (He et al., 2020a), CDNET (Raghu et al., 2021) and GPT-KE (Madotto et al., 2020).
**Explicit retrieval**: These approaches decouple the TOD system into a knowledge retriever and a response generator, including Q-TOD (Tian et al., 2022), DialoKG (Rony et al., 2022) and MAKER (Wan et al., 2023).
**Large language models**: Large language models (LLMs), such as ChatGPT (OpenAI, 2022), have demonstrated remarkable capabilities in engaging in dialogues with humans. We establish a baseline LLM utilizing ChatGPT as the response generator by leveraging the `gpt-3.5-turbo` API. To enhance its performance in TOD tasks, we integrate our knowledge retriever with the system.

## 5 Results and Analysis

In this section, we present the overall results obtained using both large-scale knowledge bases and condensed knowledge bases. Besides, we demonstrate the phenomenon of retrieval-generation misalignment and conduct ablation studies.

### 5.1 Overall Results with Large-Scale KBs

Comparing our method with others in the setting of retrieving knowledge from a large-scale knowledge base aligns better with real-world TOD scenarios. Therefore, we begin by comparing our proposed MK-TOD approach with baselines in the context of large-scale knowledge bases. The results of this comparison are displayed in Table 1. The upper section of Table 1 shows the results of methods employing a fine-tuned response generator. "Ours$_{prefix}$", "Ours$_{prompt}$", and "Ours$_{ctr}$" denote our method implementing meta knowledge using prefix, prompt, and contrastive learning techniques, respectively. "Base" and "Large" following the method names indicate the use of T5-Base or T5-Large as the response generator.

Overall, MK-TOD outperforms all previous methods with the same scale of generator model, indicating the effect of our proposed meta-knowledge. Further more, Q-TOD's retriever can achieve comparable and even higher performance than ours due to their utilization of an additional query generator. However, even when employing T5-Base and a relatively weaker retriever, our method still surpasses Q-TOD in terms of BLEU and Entity F1 by a significant margin. This indicates that our proposed method effectively utilizes the retrieved knowledge better than Q-TOD. It is also noteworthy that the Entity F1 score of T5-Large on CamRest is poorer than that of T5-Base, we attribute this to the small

| Model | MWOZ | | CamRest | | SMD | |
|---|---|---|---|---|---|---|
| | BLEU | Entity F1 | BLEU | Entity F1 | BLEU | Entity F1 |
| DF-Net (Qin et al., 2020) | 9.40 | 35.10 | - | - | 14.40 | 62.70 |
| GPT-2+KE (Madotto et al., 2020) | 15.05 | 39.58 | 18.00 | 54.85 | 17.35 | 59.78 |
| EER (He et al., 2020b) | 13.60 | 35.60 | 19.20 | 65.70 | 17.20 | 59.00 |
| FG2Seq (He et al., 2020a) | 14.60 | 36.50 | 20.20 | 66.40 | 16.80 | 61.10 |
| CDNET (Raghu et al., 2021) | 11.90 | 38.70 | 17.80 | 62.90 | 21.80 | 68.60 |
| DialoKG (Rony et al., 2022) | 12.60 | 43.50 | 23.40 | **75.60** | 20.00 | 65.90 |
| Q-TOD (T5-Base) (Tian et al., 2022) | - | - | - | - | 20.14 | 68.22 |
| Q-TOD (T5-Large) (Tian et al., 2022) | 17.62 | 50.61 | 23.75 | 74.22 | 21.33 | 71.11 |
| Ours$_{prefix}$ (Base) | 16.97 | 51.99 | 26.39 | 72.43 | 23.96 | 68.60 |
| Ours$_{prompt}$ (Base) | 17.05 | 52.42 | 25.00 | 72.09 | 23.54 | 68.28 |
| Ours$_{ctr}$ (Base) | 17.33 | 51.86 | **26.76** | 73.60 | 24.77 | 67.86 |
| Ours$_{prefix}$ (Large) | **18.02** | **53.13** | 25.68 | 71.98 | 24.97 | 72.87 |
| Ours$_{prompt}$ (Large) | 16.66 | 52.96 | 26.40 | 72.80 | 25.21 | 73.04 |
| Ours$_{ctr}$ (Large) | 17.55 | 52.97 | 26.20 | 71.72 | **25.43** | **73.31** |
| ChatGPT (OpenAI, 2022) | 7.47 | 32.87 | 15.29 | 54.71 | 14.60 | 58.11 |
| Ours$_{prefix}$ (ChatGPT) | 7.22 | 32.78 | 15.56 | 54.96 | 15.07 | 58.41 |
| Ours$_{prompt}$ (ChatGPT) | **7.58** | **35.84** | **16.07** | **56.83** | **15.24** | **59.72** |

Table 2: Overall results of E2E-TOD systems with condensed knowledge bases on MWOZ, SMD, and CamRest. The best results are highlighted in bold, and the second-best results are underlined.

size of CamRest's training data, which includes merely 406 dialogues, leading to overfitting particularly for larger models.

The bottom section of Table 2 presents the results of methods employing ChatGPT. Since ChatGPT is not fine-tunable, we do not apply contrastive learning for meta knowledge. According to the results, we note that relying solely on in-context learning does not enable ChatGPT to perform as well as the fine-tunable methods in the context of E2E-TOD. However, our proposed approach outperforms vanilla ChatGPT. Additionally, our proposed prefix method for implementing meta knowledge yields only marginal improvement or even performs worse than ChatGPT. This is attributed to ChatGPT's limited ability to learn the special prefix tokens representing meta knowledge from a limited number of in-context demonstrations and concise explanatory text. In contrast, our proposed prompt method significantly enhances its performance.

## 5.2 Overall Results with Condensed KBs

To make a comprehensive comparison with the previous methods, we also conduct evaluations on the condensed knowledge base. The results in Table 2 indicate that our proposed method surpasses the baselines on MWOZ and SMD with the same model scale, validating the efficacy of our approach. However, among the three meta knowledge implementations, it is challenging to determine a clear preference as the system with these implementations performs differently on different datasets.

For the evaluation with ChatGPT on the condensed knowledge base, we can still observe the performance gain of ChatGPT when enhanced with our proposed meta-knowledge. Besides, the performance gain is more significant than that of the large-scale knowledge bases, suggesting that ChatGPT has a higher demand for retrieval quality.

## 5.3 Retrieval-Generation Misalignment

To investigate the influence of retrieval performance on the E2E-TOD generator, we select six retrievers on MWOZ with a large-scale knowledge base. The details of the retrievers can be found in Appendix G. We then use different generators to generate responses based on the retrieval results. As for the generators, we choose Q-TOD, FiD (Izacard and Grave, 2021), and ChatGPT. The Entity F1 scores of these generators are depicted in Figures 3(a) and 3(b) as the retrieval performance varies with different retrievers.

The solid lines in the figures show that the performance of generators does not consistently align with that of retrieval performance. Furthermore, the performance of Q-TOD and FiD with oracle entities is even worse than those with a weak retriever. We refer to this phenomenon as retrieval-generation misalignment. In contrast, the dashed lines, which depict the results of the generators with our proposed meta knowledge, exhibit greater consistency between the retrieval performance and the generators. This indicates that our proposed method mitigates the misalignment issue. The correlation

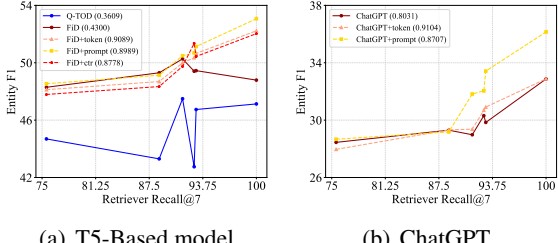

|                | (a) T5-Based model | (b) ChatGPT |

Figure 3: Entity F1 scores of generators ((a) FiD&Q-TOD and (b) ChatGPT) as the retrieval performance varies with different retrievers. Bracketed numbers following model names refer to the correlation coefficients between retrieval performance and Entity F1.

| Model | Large-Scale | | |
|---|---|---|---|
| | **BLEU** | **Entity F1** | **Recall@7** |
| Ours$_{prefix}$ | 16.39 | 50.35 | 92.51 |
| w/o MML | 16.07 | 49.56 | 91.39 |
| Ours$_{prompt}$ | 17.56 | 50.69 | 92.74 |
| w/o MML | 16.67 | 50.41 | 91.39 |
| Ours$_{ctr}$ | 15.96 | 51.35 | 92.74 |
| w/o MML | 14.78 | 50.54 | 91.39 |

Table 3: Ablation study of the MML loss.

coefficients shown in parentheses next to method names further confirm this observation.

### 5.4 Ablation Study

We assess the impact of maximum marginal likelihood, various types of meta knowledge, and the inclusion of negative samples. Unless otherwise specified, the ablation study is performed on the MWOZ dataset using T5-Base as the generator, considering the computational resource constraints.

#### 5.4.1 Maximum Marginal Likelihood

Table 3 presents the impact of the maximum marginal likelihood (MML) loss. The methods labeled as "w/o MML" utilize the warmed-up retriever described in Section 4.2, without the joint training with the response generator. The results demonstrate that the inclusion of maximum marginal likelihood enables further enhancement of the retriever during training. Consequently, the retriever leads to enhanced final responses.

#### 5.4.2 Types of Meta Knowledge

We compare different types of meta knowledge, and the results are presented in Table 4. The findings indicate that using a single type of meta knowledge yields inferior performance compared to combining all three types. Furthermore, an interesting observation emerges when using the prefix: the retrieval order outperforms other types of meta

| Method | Type | Condensed | | Large-Scale | | |
|---|---|---|---|---|---|---|
| | | **BLEU** | **Entity F1** | **BLEU** | **Entity F1** | **Recall@7** |
| Ours$_{prefix}$ | all | 16.97 | 51.99 | 16.39 | 50.35 | 92.51 |
| | order | 16.97 | 51.64 | 16.20 | 49.88 | 92.27 |
| | conf | 16.15 | 51.70 | 13.96 | 47.35 | 89.11 |
| | cooccur | 16.70 | 51.14 | 15.61 | 49.66 | 91.39 |
| Ours$_{prompt}$ | all | 17.05 | 52.42 | 17.56 | 50.69 | 92.74 |
| | order | 16.15 | 49.88 | 15.60 | 49.47 | 91.15 |
| | conf | 17.02 | 51.66 | 16.84 | 50.16 | 91.93 |
| | cooccur | 16.99 | 51.78 | 16.20 | 50.38 | 92.35 |

Table 4: Results of different types of meta knowledge on MWOZ with condensed and large-scale knowledge bases. "order", "conf", "cooccur", and "all" mean using only retrieval order, retrieval confidence, co-occurrence, and all types of meta knowledge, respectively.

| Type | Condensed | | Large-Scale | | |
|---|---|---|---|---|---|
| | **BLEU** | **Entity F1** | **BLEU** | **Entity F1** | **Recall@7** |
| Ours$_{prefix}$ (Base) | 16.97 | 51.99 | 16.39 | 50.35 | 92.51 |
| w/o neg | 16.68 | 50.45 | 15.94 | 49.46 | 90.24 |
| Ours$_{prompt}$ (Base) | 17.05 | 52.42 | 17.56 | 50.69 | 92.74 |
| w/o neg | 16.98 | 51.35 | 15.99 | 49.85 | 91.15 |
| Ours$_{ctr}$ (Base) | 17.33 | 51.86 | 15.96 | 51.35 | 92.74 |
| w/o neg | 17.11 | 50.34 | 15.79 | 48.32 | 92.29 |
| Ours$_{prefix}$ (LLM) | 7.32 | 32.38 | 6.83 | 30.47 | 92.74 |
| w/o neg | 7.22 | 32.78 | 7.01 | 30.69 | 92.74 |
| Ours$_{prompt}$ (LLM) | 7.29 | 35.98 | 6.97 | 31.88 | 92.74 |
| w/o neg | 7.58 | 36.18 | 7.31 | 32.04 | 92.74 |

Table 5: Ablation study of negative entities.

knowledge. However, when using the prompt, the results are reversed. We attribute this phenomenon to the design of the prefix and prompt. Representing meta knowledge with a prefix introduces a higher diversity in ranking order since a distinct prefix is assigned to each ranking order. This increased diversity enables the generator to better distinguish the recalled entities.

Furthermore, the distinction between retrieval confidence and co-occurrence in the prefix setting is less obvious. In contrast, when representing the meta knowledge with a prompt, the retrieval order becomes less diverse, since only the numbers representing the retrieval order are varied.

#### 5.4.3 Negative Samples

We conduct an investigation into the impact of negative entities on the performance of dialogue systems. The results presented in Table 5 demonstrate that the inclusion of negative entities significantly improves the performance of dialogue systems when applied to T5-Base. This performance enhancement can be attributed to two main factors. Firstly, the presence of negative entities facilitates easier entity distinction for the generator, enabling it to learn more effectively. Secondly, the introduction of negative entities aids in training the retriever through the MML loss in Equation (6). This concept is somewhat analogous to the moti-

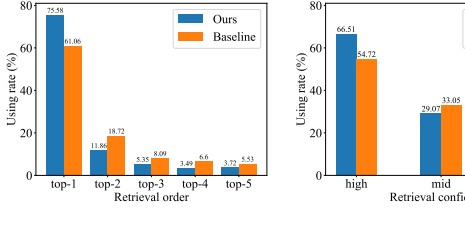

|  | |
| (a) Retrieval order | (b) Retrieval confidence |

Figure 4: The percentage of samples utilizing the entities to generate responses with respect to (a) retrieval order and (b) retrieval preference.

vation behind incorporating negative samples in knowledge retrieval tasks (Karpukhin et al., 2020).

However, when applied to ChatGPT, negative entities do not contribute to model performance. The reason is that ChatGPT cannot be fine-tuned, meaning that solely adding negative entities to the in-context demonstrations does not effectively teach ChatGPT to differentiate between entities. Consequently, we opt not to include negative entities when employing our method with ChatGPT.

### 5.5 Behavior of Generator

We examine how the generator utilizes the retrieved knowledge with the assistance of meta knowledge on the MWOZ test set. For our model and the baseline (T5-Large), we gather all their responses that contain entities and analyze the percentage of retrieved entities that appear in the responses according to retrieval order and confidence. As illustrated in Figure 4, our generator exhibits a higher propensity than the baseline to utilize entities with both a high retrieval order and high confidence. Furthermore, we have assessed the retrieval results on the MWOZ test set. The findings demonstrate that our retriever adeptly recalls 80.69% of the gold entities with a top-1 retrieval order, which directly correlates with the system-wide performance enhancement. This observation suggests that our proposed meta knowledge aids the generator in developing an inductive bias to prioritize entities that are highlighted by the retriever.

## 6 Conclusion

This paper aims to address the retrieval-generation misalignment in end-to-end task-oriented dialogue systems by introducing maximal marginal likelihood to train a perceptive retriever that leverages signals from response generation. To enable the response generator to better distinguish between entities, we explore several methods for

incorporating retrieval-related meta knowledge. We also propose to incorporate negative entities to enhance the discriminative capability. Experimental results demonstrate that when combined with meta knowledge, the response generator effectively leverages high-quality retrieval knowledge, leading to enhanced quality in the generated responses. Through analysis, we observe that previous retrieval-augmented generator models suffer from severe retrieval-generation misalignment, while our method mitigates this misalignment.

## Limitations

There are three potential limitations of the paper that warrant consideration. Firstly, the employment of the marginal likelihood method necessitates computing the likelihood for each retrieved entity, resulting in higher computational resource requirements compared to solely using negative log-likelihood (NLL). Secondly, despite conducting various comparisons and ablation studies in this paper, there are certain aspects missing in our proposed meta knowledge, such as investigating the combined utilization of prompt and contrastive learning, as well as exploring the utilization of retrieval order alongside co-occurrence. Lastly, the theoretical rationale behind the contribution of our proposed meta knowledge to task-oriented dialogue (TOD) is not thoroughly discussed.

## Acknowledgements

This work was supported by the National Natural Science Foundation of China (No. 62176270), the Guangdong Basic and Applied Basic Research Foundation (No. 2023A1515012832), and the Tencent AI Lab Rhino-Bird Focused Research Program. We thank Ke Yang for his efforts in the preliminary experiments.

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

## A  Dataset Statistics

We follow the previous work (Raghu et al., 2021) to split the datasets. The statistics of the three datasets are shown in Table 6.

| Dataset | # Dialogues Train/Val/Test | # Turns Train/Val/Test |
|---|---|---|
| MWOZ (Eric et al., 2020) | 1839/117/141 | 9943/576/711 |
| SMD (Eric et al., 2017) | 2425/302/304 | 6291/777/808 |
| CamRest (Wen et al., 2017) | 406/135/135 | 2095/675/643 |

Table 6: Statistics of the three datasets.

## B  Mapping Rules of Meta Knowledge

### B.1  Prefix of Meta Knowledge

The mapping rules from different forms of meta knowledge to the prefix tokens are shown in Table 10. We use the same set of prefix for T5 and ChatGPT. Particularly, for ChatGPT, we design a prompt paraphrase to explain the prefix to ChatGPT. This explanation prompt is shown below:

```
"Each record of knowledge base is accompanied
by three tags. The first tag indicates whether
this entity appeared before. <new-entity> means
this is a new entity, and <old-entity> means this
entity appeared before. The second tag indicates
the authenticity of the third tag. There are
three types <low-confidence>, <mid-confidence>
and <high-confidence> indicating low, middle,
high retrieval confidence respectively. A higher
retrieval confidence means the entity is
potentially more related to the user goal. The
third tag indicates its importance to the dialogue.
<nth-entity> means it is the nth important entity
in the knowledge base, for example, <1th-entity>
is the top-1 important and <other-entity> means it
is not important."
```

### B.2  Prompt of Meta Knowledge

The mapping rules from meta knowledge to the prompt are shown in Table 11. For ChatGPT, more intricate prompts are used, as presented in Table 12.

### B.3  Discussions

**How does generator learn to distinguish entities with meta knowledge?**

Our method aims to enable the models to comprehend meta knowledge, such as retrieval order and confidence scores, and then compare the retrieved entities. To achieve this, we employ distinct strategies based on whether the models are fine-tunable

or not. For fine-tunable models like T5, we include the meta knowledge as a part of the model input, *enabling its assimilation during the training process*. In the case of non-fine-tunable models like ChatGPT, our approach *utilizes the model's existing knowledge to grasp the essence of meta knowledge*. This is implemented by providing an explanation of meta knowledge within the system prompt, as elaborated in Appendix B.1 and Appendix B.2.

**How to design the prompts of meta knowledge?** When it comes to the prompts for T5 models, given their fine-tunability, our emphasis lies in the effectiveness of implementation with the `pytorch` code, and leave the comprehension of prompts to the model itself during the training phase. To achieve this, we craft the prompts in a manner that permits them to be tokenized into sequences of uniform length with the `tokenizer`, which facilitates efficient mapping to entities through the `torch.gather` operation.

As for ChatGPT, the final prompts are obtained through evaluations on the dev set of MWOZ.

## C In-Context Learning Demonstration

The inputs for ChatGPT to include meta knowledge by the prefix and prompt approaches are shown in Figure 5 and Figure 6, respectively.

## D Pre-training for Retriever

To pre-train the retriever, we employ a distant supervision method. This involves labeling the entity that exhibits the highest occurrence of attribute values in both the dialogue context and system response as the pseudo positive entity. Subsequently, we conduct pre-training of the retriever using in-batch contrastive learning, considering the positive entities from other examples within the same mini-batch as negative entities.

## E Hyperparameter Settings

The hyperparameters of our system with both condensed and large-scale knowledge bases are shown in Table 13. We also retrieve different numbers of entities for different datasets in our experiments, as the details shown in Table 7.

## F Result of MK-TOD with T5-Large on Large-Scale Knowledge Bases

The results of our method built upon T5-Large are shown in Table 8.

| | Condensed KB | | | Large-Scale KB | |
| | MWOZ | CamRest | SMD | MWOZ | CamRest |
|---|---|---|---|---|---|
| KB size | 7 | 7 | 8 | 223 | 112 |
| # Retrieved entities for T5-Base | 6 | 6 | 8 | 7 | 7 |
| # Retrieved entities for T5-Large | 5 | 5 | - | 5 | 5 |

Table 7: The number of retrieved entities under different settings.

| Model | MWOZ | | | CamRest | | |
| | BLEU | Entity F1 | Recall@5 | BLEU | Entity F1 | Recall@5 |
|---|---|---|---|---|---|---|
| Ours$_{prefix}$ (Large) | 16.69 | 53.59 | 84.58 | 27.32 | 72.77 | 87.05 |
| Ours$_{prompt}$ (Large) | 17.15 | 52.99 | 88.66 | 26.88 | 72.92 | 92.94 |
| Ours$_{ctr}$ (Large) | 17.40 | 53.26 | 93.19 | 27.82 | 71.98 | 92.70 |

Table 8: Overall results of E2E-TOD systems with large-scale knowledge bases on MWOZ and CamRest.

## G Details of Retrievers for Section 5.3

In Section 5.3, we investigated the retrieval-generation misalignment by introducing several retrievers with different performances. The details of these retrievers are introduced as follows.

**BM25:** The BM25 retriever computes the BM25 score between the dialogue context and each entity.

**Frequency:** This is a rule-based method. For each entity in the knowledge base, we compute the number of its attribute values that appear in the context. We then take the entities with the most attribute values appearing in the dialogue context as the recalled entities.

**Pre-train:** This retriever is the pre-trained retriever introduced in Appendix D.

**Ours:** This is the retriever introduced in our method (Ours$_{prompt}$(Base)).

**Q-TOD:** This is the retriever of Q-TOD.

**Oracle:** This method uses the condensed knowledge base as the retrieved entity set. The gold entity, which must appear in the condensed knowledge base, is marked as the top-1 retrieved entity with high retrieval confidence, while other entities are marked with low retrieval confidence.

We show the performance (Recall@7) of these retrievers in Table 9.

| BM25 | Frequency | Pre-train | Ours | Q-TOD | Oracle |
|---|---|---|---|---|---|
| 75.51 | 88.66 | 91.39 | 92.74 | 92.97 | 100 |

Table 9: The performance (Recall@7) of different retrievers for the retrieval-generation misalignment study.

| Meta Knowledge | Prefix |
|---|---|
| **Retrieval Order** | |
| Firstly recalled entity | `<1th-entity>` |
| Secondly recalled entity | `<2th-entity>` |
| Thirdly recalled entity | `<3th-entity>` |
| Fourthly recalled entity | `<4th-entity>` |
| Fifthly recalled entity | `<5th-entity>` |
| The entities recalled behind the 5th entity and the easy negative entity | `<other-entity>` |
| **Retrieval Confidence** | |
| Entity with retrieval score >= 0.75 | `<high-confidence>` |
| Entity with retrieval score < 0.75 and >= 0.25 | `<mid-confidence>` |
| Entity with retrieval score < 0.25 and the easy negative entity | `<low-confidence>` |
| **Co-occurrence Relation** | |
| Entity existed in the dialogue context | `<old-entity>` |
| Entity not existed in the dialoglue context and the easy negative entity | `<new-entity>` |

Table 10: The mapping rules from different types of meta knowledge to the prefix token.

| Meta Knowledge | Prompt |
|---|---|
| **Retrieval Order** | |
| Firstly recalled entity | "The top-1 recalled:" |
| Secondly recalled entity | "The top-2 recalled:" |
| Thirdly recalled entity | "The top-3 recalled:" |
| Fourthly recalled entity | "The top-4 recalled:" |
| Fifthly recalled entity | "The top-5 recalled:" |
| The entities recalled behind the 5th entity and the easy negative entity | "The negative entity recalled:" |
| **Retrieval Confidence** | |
| Entity with retrieval score >= 0.75 | "with high confidence:" |
| Entity with retrieval score < 0.75 and >= 0.25 | "with middle confidence:" |
| Entity with retrieval score < 0.25 and the easy negative entity | "with low confidence:" |
| **Co-occurrence Relation** | |
| Entity existed in the dialogue context | "existed in history:" |
| Entity not existed in the dialoglue context and the easy negative entity | "newly recalled:" |

Table 11: The mapping rules from different types of meta knowledge to the prompt for T5.

| Meta Knowledge | Prompt |
|---|---|
| **Retrieval Order** | |
| Firstly recalled entity | "this entity is top-1 important." |
| Secondly recalled entity | "this entity is top-2 important." |
| Thirdly recalled entity | "this entity is top-3 important." |
| Fourthly recalled entity | "this entity is top-4 important." |
| Fifthly recalled entity | "this entity is top-5 important." |
| The entities recalled behind the 5th entity and the easy negative entity | "this entity is not important." |
| **Retrieval Confidence** | |
| Entity with retrieval score >= 0.75 | "It has high possibility that" |
| Entity with retrieval score < 0.75 and >= 0.25 | "It has medium possibility that" |
| Entity with retrieval score < 0.25 and the easy negative entity | "It has low possibility that" |
| **Co-occurrence Relation** | |
| Entity existed in the dialogue context | "This entity has appeared before." |
| Entity not existed in the dialoglue context and the easy negative entity | "This is a new entity." |

Table 12: The mapping rules from different types of meta knowledge to the prompt for ChatGPT.

| Hyperparameters | Condensed KB | | Large-Scale KB | |
| --- | --- | --- | --- | --- |
| | **T5-Base** | **T5-Large** | **T5-Base** | **T5-Large** |
| Batch size | 2 | 1 | 2 | 1 |
| Gradient accumulation steps | 32 | 64 | 32 | 64 |
| Training gradient steps | | 1500 | | |
| Learning rate schedule | | Linear | | |
| Retriever learning rate | | 1e-4 | | |
| Response generator learning rate | | 1e-4 | | |
| Gradient weight decay | | 0.01 | | |
| Gradient clipping | | 0.01 | | |
| Retriever max input length | | 128 | | |
| Generator max input length for context | | 200 | | |
| Generator max input length for entity | | 100 | | |
| Max output length | | 64 | | |
| Loss weight $\alpha$ for Eq. 7 | | 1 | | |
| Loss weight $\beta$ for Eq. 7 | | 1 | | |
| Loss weight $\gamma$ for Eq. 11 | | 1 | | |
| Margin $\lambda$ for contrastive learning in Eq. 10 | | 0.01 | | |

Table 13: Hyperparameter settings of our system.

```
You answer questions like a customer service. There is a knowledge base for each question.
Each record of knowledge base is accompanied by three tags.
The first tag indicates whether this entity appeared before. <new-entity> means this is a
new entity, and <old-entity> means this entity appeared before.
The second tag indicates the authenticity of the third tag. There are three types <low-
confidence>, <mid-confidence> and <high-confidence> indicating low, middle, high retrieval
confidence respectively. Higher retrieval confidence mean the entity is potentially more
related to the user goal.
The third tag indicates its importance to the dialogue. <nth-entity> means it is the nth
important entity in the knowledge base, for example, <1th-entity> is the top-1 important and
<other-entity> means it is not important.
The max length of your response is 50 words.

Next are some demonstrations.
There are three special tokens [assistant], [user] and [answer]. [assistant] leads the
response of the customer service, [user] leads what user say and [answer] leads the Ground
Truth answer of the example.

You answer questions like a customer service of hotel reservation with a knowledge base.
Knowledge base is in the form of : address | area | internet | name | parking | phone |
postcode | pricerange | stars | type.  Knowledge base is as follow.

First one : 124 tenison road | east | yes | a and b guest house | 01223315702 | cb12dp |
moderate | 4 star | guesthouse  <1st-entity> <low-confidence> <new-entity>
Next one : ......
Next one : ......

That's all of knowledge base. The dialogue is as follow:
[user] yes , i am looking for a place to stay tonight . the hotel should be like a
guesthouse in looks and style . ideally , i ' d like one in the moderate price range ,
please .
[answer] is there a specific area you would like to stay in ? also , do you need internet
and / or free parking ?
That's all of example 1
......

{Add the test sample here}
```

Figure 5: The input prompt and demonstration for ChatGPT with meta knowledge as the prefix.

```
You answer questions like a customer service. There is a knowledge base for each question.
The max length of your response is 50 words.

Next are some demonstrations.
There are three special tokens [assistant], [user] and [answer]. [assistant] leads the
response of the customer service, [user] leads what user say and [answer] leads the Ground
Truth answer of the example.

You answer questions like a customer service of hotel reservation with a knowledge base.
Knowledge base is in the form of : address | area | internet | name | parking | phone |
postcode | pricerange | stars | type.  Knowledge base is as follow.

First one : 124 tenison road | east | yes | a and b guest house | 01223315702 | cb12dp |
moderate | 4 star | guesthouse  This is a new entity. It has low possibility that this
entity is top-1 important.
Next one : ......
Next one : ......

That's all of knowledge base. The dialogue is as follow:
[user] yes , i am looking for a place to stay tonight . the hotel should be like a
guesthouse in looks and style . ideally , i ' d like one in the moderate price range ,
please .
[answer] is there a specific area you would like to stay in ? also , do you need internet
and / or free parking ?
That's all of example 1
......

{Add the test sample here}
```

Figure 6: The input prompt and demonstration for ChatGPT with meta knowledge as prompt.