# OpenReview forum: "Retrieval-Generation Alignment for End-to-End Task-Oriented Dialogue System"
_EMNLP/2023/Conference — EMNLP 2023 Main_

### Official Review · Reviewer_ybw6 · 2023-08-04

**Typos Grammar Style And Presentation Improvements:** Meta knowledge is misspelled as mata …
**Soundness:** 4

**Excitement:**

3: Ambivalent: It has merits (e.g., it reports state-of-the-art results, the idea is nice), but there are key weaknesses (e.g., it describes incremental work), and it can significantly benefit from another round of revision. However, I won't object to accepting it if my co-reviewers champion it.

**Paper Topic And Main Contributions:**

The paper raised a "misalignment" problem in E2E-TOD systems, where the generator may sometimes mismatch with the retriever, so that better performance of the retriever may not lead to a better performance of the whole system. To alleviate the problem, the paper introduces a novel framework where their so-called meta-knowledge (the confidence of the retriever etc.) is taken into consideration. Multiple methods including special prefix tokens, prompts and contrastive learning are tried for the usage of the proposed framework, and meta-knowledge proves to work for various systems.

**Reasons To Accept:**

The paper puts forward the misaligned problem in the retrieval-generation process, which is of practical significance. The meta-knowledge they introduce in the process combines the retriever and generator more tightly and proves to work for both small and large models, which means the meta-knowledge might be introduced in various future works in similar paradigms.

**Reasons To Reject:**

1. The scores chatgpt reaches in the experiment are too low whether meta-knowledge is introduced, which means other metrics other than BLEU and Entity F1 might be necessary for the evaluation.
2. More types of meta-knowledge and more methods to leverage them might be applied in the framework.
3. The consistence between the quality of meta-knowledge, the performance retriever and ground-truth lacks study. It’s better to include more analysis in the future stage.


**Reproducibility:**

4: Could mostly reproduce the results, but there may be some variation because of sample variance or minor variations in their interpretation of the protocol or method.

**Reviewer Confidence:**

4: Quite sure. I tried to check the important points carefully. It's unlikely, though conceivable, that I missed something that should affect my ratings.

---

> ### Author Rebuttal · Authors · 2023-08-28
>
> ***Q1: The scores chatgpt reaches in the experiment are too low whether meta-knowledge is introduced, which means other metrics other than BLEU and Entity F1 might be necessary for the evaluation.***
>
> A1: Despite the application of in-context learning, ChatGPT continues to attain relatively lower Entity F1 scores, mainly because it hasn't undergone fine-tuning on the E2E-TOD datasets. However, this shouldn't be interpreted as ChatGPT performing inadequately on these datasets. Certain behaviors like generating overly lengthy responses or including unnecessary attributes may account for the lower scores. We will present the results of human evaluation once it's completed.
>
> ***Q2: More types of meta-knowledge and more methods to leverage them might be applied in the framework.***
>
> A2: Indeed, our framework can accommodate a broader range of meta-knowledge types. We will further investigate these possibilities using widely adopted fine-tunable language models, including Llama-2-chat.
>
>
> ***Q3: The consistence between the quality of meta-knowledge, the performance retriever and ground-truth lacks study. It’s better to include more analysis in the future stage.***
>
> A3: We appreciate your suggestions. We have assessed the retrieval results on the MWOZ test set. The findings demonstrate that our retriever adeptly recalls 80.69% of the gold entities with a top-1 retrieval order. This directly correlates with the system-wide performance enhancement, as shown in Section 5.5, where our generator prioritizes entities with higher retrieval orders. We will include the result when updating the paper.
>
> We appreciate your positive feedback. Please let us know if you have any further questions, and we are happy to discuss further.

---

### Official Review · Reviewer_YzME · 2023-08-05

**Soundness:** 4

**Excitement:**

3: Ambivalent: It has merits (e.g., it reports state-of-the-art results, the idea is nice), but there are key weaknesses (e.g., it describes incremental work), and it can significantly benefit from another round of revision. However, I won't object to accepting it if my co-reviewers champion it.

**Paper Topic And Main Contributions:**

This paper addresses shortcomings in retrieve-and-generate approaches for knowledge-grounded dialogue systems, and specifically aims to improve the information flow from the retriever output to the generator, as they observe that improvements in retrieval performance often do not correlate with improvements in generation. To that end, the paper proposes to include 'meta-knowledge' labels that denote the retrieval order, confidence, and entity co-occurrence, alongside the input of the generator. The retriever is trained by maximizing the marginal likelihood of the response and the generator is further trained using contrastive learning and negative entity sampling (with corresponding 'meta-knowledge' labels).

The paper presents results on three datasets (MultiWoz, CamRest, SMD) and on two settings: using session-level or dataset-level knowledge bases. The proposed additions lead to consistently better output quality (measured by BLEU) but the performance increase is not as prevalent on Entity-F1 when using session-level knowledge bases, and Recall@7 when using dataset-level  knowledge bases. The paper also includes results when employing ChatGPT as the generator and prompting it with the retriever output. The included meta-knowledge show consistent improvements across the board in this setting, but cannot be compared with previous work.

The paper also includes some analysis that shows that when using their additions, the correlation between retriever and generator performance increases, as well as ablation experiments over how to best represent the additional knowledge (prefix vs. prompt), the use of maximum marginal likelihood, the effect of including different types of "meta-knowledge" labels, and negative samples.


**Questions For The Authors:**

A. Did the authors explore using the raw retrieval order and confidence scores? On the same topic, did the authors explore a more granular mapping of the confidence scores?

B. On Table 1, why are results not presented for the Ours_x(Large) settings on SMD?

C. In 5.4.1, the "w/o MML" settings mean that the pre-trained retriever is not further trained at all?

D. Looking at tables 8 and 9, what is the motivation between different prompts being used in T5 vs. ChatGPT? Have the authors explored using other prompting language, and how has that affected performance?


**Reasons To Accept:**

- The proposed method is simple to understand and is shown to lead to consistent performance gains in a variety of settings.
- The paper includes thorough analysis and properly supports the efficacy of its proposed approach.
- Well motivated and written paper.


**Reasons To Reject:**

- The mapping of retrieval order and confidence scores (both continuous numerical values) to discrete and independent string labels seems arbitrary and dismisses how the retrieved entities may compare to each other.
- The prompt based results may be particular to the specific language used to create the prompts. The paper includes no study on alternative phrasings of the prompt.


**Reproducibility:**

4: Could mostly reproduce the results, but there may be some variation because of sample variance or minor variations in their interpretation of the protocol or method.

**Reviewer Confidence:**

4: Quite sure. I tried to check the important points carefully. It's unlikely, though conceivable, that I missed something that should affect my ratings.

**Typos Grammar Style And Presentation Improvements:**

- The conclusions of section 5.4.2 are a bit confusing ('on the other hand' followed by 'in contrast'), please rephrase.

---

> ### Author Rebuttal · Authors · 2023-08-28
>
> ***Q1: The mapping of retrieval order and confidence scores (both continuous numerical values) to discrete and independent string labels seems arbitrary and dismisses how the retrieved entities may compare to each other.***
>
> A1: Good question. Our method aims to enable the models to comprehend meta knowledge, such as retrieval order and confidence scores, and then compare the retrieved entities. To achieve this, we employ distinct strategies based on whether the models are fine-tunable or not. For fine-tunable models like T5, we include the meta knowledge as a part of the model input, enabling its assimilation during the training process. In the case of non-fine-tunable models like ChatGPT, our approach utilizes the model's existing knowledge to grasp the essence of meta knowledge. This is implemented by providing an explanation of meta knowledge within the system prompt, as elaborated in Appendix B.
>
> ***Q2: The prompt based results may be particular to the specific language used to create the prompts. The paper includes no study on alternative phrasings of the prompt.***
>
> A2: We agree. We have experimented with various prompts to arrive at the final one, particularly for ChatGPT. Due to the limit of space, we were unable to thoroughly showcase the prompt design process. We will provide a detailed introduction of the prompt design process in our future revisions.
>
> ***Q3: Did the authors explore using the raw retrieval order and confidence scores? On the same topic, did the authors explore a more granular mapping of the confidence scores?***
>
> A3: Good question. Regarding the ranking order, which involves discrete numbers, we incorporate the actual raw ranking number directly into the prompt. For instance, the number "1" is included in the prompt "the top 1 recalled." However, as evidenced by the results in Table 4 (Ours_prompt (order)), T5 exhibits limitations in comprehending the order conveyed by these discrete numbers.
>
> In terms of confidence scores, which entail continuous numbers, multiple studies have demonstrated that even powerful models like ChatGPT struggle with precise numerical problem-solving (https://arxiv.org/abs/2103.07191). Consequently, in this paper, we opt for a course-grained confidence measure rather than relying on exact confidence scores.
>
> ***Q4: On Table 1, why are results not presented for the Ours_x(Large) settings on SMD?***
>
> A4: When conducting this research, our experiments were ran using a 24G 3090 GPU, which fell short of the requirements for the SMD dataset, with its demand for longer model input length. Currently, we have finished the experiments on the SMD dataset using T5-large with 40g A100 GPUs, and the results are presented below. We will provide the missing results when updating the paper.
>
> | Methods          | BLEU  | Entity F1 |
> | ------------------- | ----- | --------- |
> | Ours_prefix (Large) | 24.97 | 72.87     |
> | Ours_prompt (Large) | 25.21 | 73.04     |
> | Ours_ctr (Large)    | 25.43 | 73.31     |
>
> ***Q5:. In 5.4.1, the "w/o MML" settings mean that the pre-trained retriever is not further trained at all?***
>
> A5: Yes, we will provide explicit clarification on this.
>
> ***Q6: Looking at tables 8 and 9, what is the motivation between different prompts being used in T5 vs. ChatGPT? Have the authors explored using other prompting language, and how has that affected performance?***
>
> A6: When it comes to prompts for T5 models, given their fine-tuning capability, our emphasis lies in the effectiveness of implementation with the ```pytorch``` code, and we leave the comprehension of prompts to the model itself during the training phrase. To achieve this, we craft the prompts in a manner that permits them to be tokenized into sequences of uniform length with the ```tokenizer```, which facilitate the efficient mapping to entities through the ```torch.gather``` operation.
>
> For prompts for ChatGPT, we have experimented with various prompts to arrive at the final one. Due to the limit of space, we were unable to thoroughly showcase the prompt design process. We will provide a detailed introduction of the prompt design process in our future revisions.
>
> Thank you for the review! Feel free to let us know if you have more questions.

---

### Official Review · Reviewer_b5KD · 2023-08-07

**Soundness:** 4

**Excitement:**

3: Ambivalent: It has merits (e.g., it reports state-of-the-art results, the idea is nice), but there are key weaknesses (e.g., it describes incremental work), and it can significantly benefit from another round of revision. However, I won't object to accepting it if my co-reviewers champion it.

**Paper Topic And Main Contributions:**

The authors propose MK-TOD to address the retrieval-generation misalignment in the end-to-end task-oriented dialogue system. The authors train the retriever and the generator in the end-to-end dialogue system and apply MML from the response generation as the signal. MK-TOD considers both retrieved entities and meta knowledge to guide the generator and the results show better performance when combined with meta knowledge.

**Questions For The Authors:**

A: What are the missing results in Table 1 for T5-Large? Have you analyzed why the base model has better performance on CamRest?

**Reasons To Accept:**

- Meta knowledge for retrieval is easy to use and the result shows marginal improvement in the ablation study.

- The retriever and the generator are trained in an end-to-end manner and apply the response generation as the signal to update the retriever.

- Authors explore different techniques further improve the performance including contrastive learning, negative entity

**Reasons To Reject:**

- Automatic evaluation is not enough. Human evaluation should be included in the generation task.

- There are some missing results in Table1 for T5-Large on SMD dataset. It seems that the experiments have not been finished yet. Thus, the results do not support the claim that the proposed method surpasses the baselines on MWOZ and SMD (Line429-432). It is also strange to see that the large model provides poorer performance than the Base model on CamRest.

- In this work, Entity F1 is the key metric to evaluate the retrieval-generation misalignment. Table 2, which should be highlighted, shows good performance on large-scale knowledge bases. Table 1, the main table, cannot convince me that the proposed method improves the performance since only results on MWOZ provide better Entity F1.

**Reproducibility:**

3: Could reproduce the results with some difficulty. The settings of parameters are underspecified or subjectively determined; the training/evaluation data are not widely available.

**Reviewer Confidence:**

3: Pretty sure, but there's a chance I missed something. Although I have a good feel for this area in general, I did not carefully check the paper's details, e.g., the math, experimental design, or novelty.

**Typos Grammar Style And Presentation Improvements:**

- Highlight Table 2 instead of Table 1

---

> ### Author Rebuttal · Authors · 2023-08-28
>
> ***Q1: Automatic evaluation is not enough. Human evaluation should be included in the generation task.***
>
> A1: Thanks for the suggestion. We are conducting a human evaluation and will report the result as soon as it is finished.
>
> ***Q2: There are some missing results in Table1 for T5-Large on SMD dataset. It seems that the experiments have not been finished yet. Thus, the results do not support the claim that the proposed method surpasses the baselines on MWOZ and SMD (Lines 429-432). It is also strange to see that the large model provides poorer performance than the Base model on CamRest.***
>
> A2: There are no results for T5-large on the SMD dataset because we only used a single 24G NVIDIA RTX 3090 GPU. However, this didn't meet the SMD dataset's requirements, which needed a longer model input length. Currently, we have added to the experiments on the SMD dataset by using T5-large with 40g A100 GPUs, and we're presenting the results below. We'll include the missing results when we update the paper.
> |Methods           | BLEU  | Entity F1 |
> | ------------------- | ----- | --------- |
> | Ours_prefix (Large) | 24.97 | 72.87     |
> | Ours_prompt (Large) | 25.21 | 73.04     |
> | Ours_ctr (Large)    | 25.43 | 73.31     |
>
> The reason behind the weaker performance of T5-large compared to T5-base on the Camrest dataset can be traced back to the small size of its training data. This dataset includes merely 406 dialogues, comprising a total of 2095 responses, which potentially leads to overfitting particularly for larger models such as T5-large.
>
>
> ***Q3: In this work, Entity F1 is the key metric to evaluate the retrieval-generation misalignment. Table 2, which should be highlighted, shows good performance on large-scale knowledge bases. Table 1, the main table, cannot convince me that the proposed method improves the performance since only results on MWOZ provide better Entity F1.***
>
> A3: We agree that the emphasis should be placed on the results in Table 2. The findings depicted in Table 1 primarily showcase the application of a concise knowledge base (containing only 7-8 entities), which doesn't effectively reflect the impact of retrieval-generation alignment. We will revise this in the forthcoming revision.
>
> Thank you once again for your review! We are more than happy to engage in further discussions.

---

### Official Review · Reviewer_myc6 · 2023-08-10

**Soundness:** 3

**Excitement:**

2: Mediocre: This paper makes marginal contributions (vs non-contemporaneous work), so I would rather not see it in the conference.

**Paper Topic And Main Contributions:**

In a retrieval-augmented TOD, the correlation between the performance of the retriever and the generator is weak. Thus, the improvements in the retriever may not translate to enhancing the generator.

To solve the above problem, the authors propose MK-TOD. It contains the maximum marginal likelihood to assist the retriever to train synchronously with the generator, and the meta-knowledge of entities to help the generator utilize the most appropriate entity among all the retrieved ones.
The experiments demonstrate that MK-TOD improves the correlation between the performance of the retriever and the generator, and makes a certain improvement in generation and retrieval.

**Reasons To Accept:**

1. The problem this paper focuses on is of interest.
Previous works mainly focus on improving retrieval and generation performance separately, while ignoring how to make the improvement of retrieval better serve the final generation performance.

2. This paper clearly states the motivations of the authors and the implementation of the method, which brings a low reading burden.

**Reasons To Reject:**

1. The experiments do not demonstrate the effectiveness of the proposed MK-TOD.
a)  In Table 1, the MK-TOD fails to outperform baselines in Entity F1 on the two datasets (CamRest & SMD).
b)  In Table 1, the authors do not show the performance of the MK-TOD-Large on SMD.


2. The proposed MK-TOD is of limited innovation.
The essence of MK-TOD is to simply make the generator focus on the top-ranked entities in the retrieval ones. The idea is not refreshing.

3. The paper lacks validation for numerous hyperparameters. So the proposed method seems to use some tricks. For example, in Table 10, the loss weight in Equ.7 and Equ.11 lacks explanation and validation. In line 269, the authors do not explain and validate the thresholds for categorizing entities.

**Reproducibility:**

4: Could mostly reproduce the results, but there may be some variation because of sample variance or minor variations in their interpretation of the protocol or method.

**Reviewer Confidence:**

4: Quite sure. I tried to check the important points carefully. It's unlikely, though conceivable, that I missed something that should affect my ratings.

---

> ### Author Rebuttal · Authors · 2023-08-28
>
> ***Q1: The experiments do not demonstrate the effectiveness of the proposed MK-TOD. a) In Table 1, the MK-TOD fails to outperform baselines in Entity F1 on the two datasets (CamRest & SMD). b) In Table 1, the authors do not show the performance of the MK-TOD-Large on SMD.***
>
> A1: In Table 1, our focus is directed towards the comparison of methods with identical model sizes, such as QTOD (T5-base) v.s. ours (Base). Drawing from this criterion, we claim that our proposed method demonstrates superior performance over the baseline methods on both MWOZ and SMD datasets.
>
> Regarding the missing results for T5-large on the SMD dataset, we conducted experiments using only a single 24G NVIDIA RTX 3090 GPU. However, this fell short of what the SMD dataset needed in terms of longer model input. Now, we've included the experiments on the SMD dataset using T5-large with 40g A100 GPUs, and we've displayed the results below. As we update the paper, we'll also supply the missing results.
>
> | Methods          | BLEU  | Entity F1 |
> | ------------------- | ----- | --------- |
> | Ours_prefix (Large) | 24.97 | 72.87     |
> | Ours_prompt (Large) | 25.21 | 73.04     |
> | Ours_ctr (Large)    | 25.43 | 73.31     |
>
> Furthermore, it's important to acknowledge that the outcomes displayed in Table 1 provide an incomplete depiction of our method's performance. As elaborated in Lines 351-360 and detailed in Section 5.2, each dialogue within Table 1 corresponds to a restricted set of 7-8 entities, thereby diminishing the emphasis on retrieval. **In light of this, we contend that Table 2 deserves greater consideration for evaluating our results.** Responding to the feedback from Reviewer b5KD, we intend to emphasize Table 2 in stead of Table 1 in order to avert any potential misunderstandings in the upcoming revision.
>
> ***Q2: The proposed MK-TOD is of limited innovation. The essence of MK-TOD is to simply make the generator focus on the top-ranked entities in the retrieval ones. The idea is not refreshing.***
>
> A2: Besides the ranking order, we also introduce confidence and co-occurrence as meta-knowledge to enhance the retrieval-generation alignment of E2E-TOD. At the same time, we introduce prefix, prompt, and contrastive learning to implement these components. We extensively evaluate the above components in Section 5.4.2. Besides, except for the meta-knowledge, we also introduce negative sample in training the response generator, which is novel and effective, as validated in Table 5. Notably, our work is the first attempt to include meta knowledge in E2E-TOD, providing valuable insights for future research in this field.
>
> ***Q3: So the proposed method seems to use some tricks. For example, in Table 10, the loss weight in Equ.7 and Equ.11 lacks explanation and validation. In line 269, the authors do not explain and validate the thresholds for categorizing entities.***
>
> A3: Due to limitations in computational resources, we didn't extensively fine-tune hyperparameters. Instead, we opted for a balanced 1:1:1 ratio for the loss weights in Eq. 7 and Eq. 11. We recognize that we didn't thoroughly discuss the thresholds for categorizing entities. To provide clarity, we established the following ranges: (-infinity, 0.4] indicates low confidence, (0.4, 0.75] indicates medium confidence, and (0.75, +infinity) indicates high confidence. These thresholds were chosen based on an empirical evaluation of the retrieval scores. We will incorporate these clarifications when updating the paper.
>
> Thanks again for your review! Please let us know if you have any further questions, and we are happy to discuss further.

---

### Meta-Review · Area_Chair_fSGd · 2023-09-19

**Recommendation:** 4

**Metareview:**

Reviewers agree that the work presented in the paper is well-motivated and paper is generally well-written. Empirically, the proposed method achieves marginal improvements. Questions around missing experiments and result analysis were largely addressed during rebuttal.

---

### Decision · Program_Chairs · 2023-10-07

**Decision:**

Accept-Main

**Comment:**

Reviewers agree that the work presented in the paper is well-motivated and paper is generally well-written. Empirically, the proposed method achieves marginal improvements. Questions around missing experiments and result analysis were largely addressed during rebuttal.